# A Prognostic Model Using Post-Steroid Neutrophil-Lymphocyte Ratio Predicts Overall Survival in Primary Central Nervous System Lymphoma

**DOI:** 10.3390/cancers14071818

**Published:** 2022-04-03

**Authors:** Yu Tung Lo, Vivian Yujing Lim, Melissa Ng, Ya Hwee Tan, Jianbang Chiang, Esther Wei Yin Chang, Jason Yongsheng Chan, Eileen Yi Ling Poon, Nagavalli Somasundaram, Mohamad Farid Bin Harunal Rashid, Miriam Tao, Soon Thye Lim, Valerie Shiwen Yang

**Affiliations:** 1Department of Neurosurgery, National Neuroscience Institute, 11 Jalan Tan Tock Seng, Singapore 308433, Singapore; yutung.lo@mohh.com.sg; 2Department of Neurosurgery, Singapore General Hospital, Outram Road, Singapore 169608, Singapore; 3Translational Precision Oncology Lab, Institute of Molecular and Cell Biology (IMCB), A*STAR, 61 Biopolis Dr, Proteos, Singapore 138673, Singapore; limvyj@imcb.a-star.edu.sg; 4Singapore Immunology Network (SIgN), A*STAR, 8A Biomedical Grove, Immunos, Singapore 138648, Singapore; melissa_ng@immunol.a-star.edu.sg; 5Division of Medical Oncology, National Cancer Centre Singapore, 11 Hospital Crescent, Singapore 169610, Singapore; tan.ya.hwee@singhealth.com.sg (Y.H.T.); chiang.jianbang@singhealth.com.sg (J.C.); esther.chang.w.y@singhealth.com.sg (E.W.Y.C.); jason.chan.y.s@singhealth.com.sg (J.Y.C.); eileen.poon.y.l@singhealth.com.sg (E.Y.L.P.); nagavalli.somasundaram@singhealth.com.sg (N.S.); mohamad.farid@singhealth.com.sg (M.F.B.H.R.); miriam.tao@singhealth.com.sg (M.T.); lim.soon.thye@singhealth.com.sg (S.T.L.); 6Oncology Academic Clinical Program, Duke-NUS Medical School, 8 College Road, Singapore 169857, Singapore

**Keywords:** PCNSL, lymphoma, hematological index, nlr, prognosis

## Abstract

**Simple Summary:**

Hematological indices such as neutrophil-lymphocyte ratio (NLR) have been found to be prognostic for survival outcomes, with higher NLR portending a worse prognosis in primary central nervous system lymphomas (PCNSLs) and other cancers. However, corticosteroids, commonly used for reducing cerebral edema, as well as being a part of systemic treatment, subsequently alter the balance of neutrophil and lymphocyte composition in the peripheral circulation. We hypothesized that the response to corticosteroids may correlate with the response of PCNSL to systemic treatment and survival. We, therefore, investigated the NLR before and after steroids, and found that higher post-steroid NLR was paradoxically correlated with better survival. We thus developed a new decision-tree-based prognostic score using age, post-steroid NLR and pre-steroid NLR, and showed that it stratified patients into three risk profiles that predicted overall survival with good discrimination and calibration in patient cohorts across two different centers.

**Abstract:**

Background: Ratios of differential blood counts (hematological indices, HIs) had been identified as prognostic variables in various cancers. In primary central nervous system lymphomas (PCNSLs), higher baseline neutrophil-lymphocyte ratio (NLR) in particular was found to portend a worse overall survival. However, it was often observed that differential counts shift drastically following steroid administration. Moreover, steroids are an important part of the arsenal against PCNSL due to its potent lymphotoxic effects. We showed that the effect of steroids on differential blood cell counts and HIs could be an early biomarker for subsequent progression-free (PFS) and overall survival (OS). Methods: This study retrospectively identified all adult patients who received a brain biopsy from 2008 to 2019 and had histologically confirmed PCNSL, and included only those who received chemoimmunotherapy, with documented use of corticosteroids prior to treatment induction. Different blood cell counts and HIs were calculated at three time-points: baseline (pre steroid), pre chemoimmunotherapy (post steroid) and post chemoimmunotherapy. Tumor progression and survival data were collected and analyzed through Kaplan–Meier estimates and Cox regression. We then utilized selected variables found to be significant on Kaplan–Meier analysis to generate a decision-tree prognostic model, the NNI-NCCS score. Results: A total of 75 patients who received chemoimmunotherapy were included in the final analysis. For NLR, OS was longer with higher pre-chemoimmunotherapy (post-steroid) NLR (dichotomized at NLR ≥ 4.0, HR 0.42, 95% CI: 0.21–0.83, *p* = 0.01) only. For platelet-lymphocyte ratio (PLR) and lymphocyte-monocyte ratio (LMR), OS was better for lower post-chemoimmunotherapy PLR (dichotomized at PLR ≥ 241, HR 2.27, 95% CI: 1.00 to 5.18, *p* = 0.05) and lower pre-chemoimmunotherapy (post-steroid) LMR (dichotomized at LMR ≥25.7, HR 2.17, 95% CI: 1.10 to 4.31, *p* = 0.03), respectively, only. The decision-tree model using age ≤70, post-steroid NLR >4.0, and pre-steroid (baseline) NLR <2.5 and the division of patients into three risk profiles—low, medium, and high—achieved good accuracy (area-under-curve of 0.78), with good calibration (Brier score: 0.16) for predicting 2-year overall survival. Conclusion: We found that post-steroid NLR, when considered together with baseline NLR, has prognostic value, and incorporation into a prognostic model allowed for accurate and well-calibrated stratification into three risk groups.

## 1. Introduction

Neutrophil-lymphocyte ratio (NLR) is a simple, easily available and inexpensive prognostic marker in many different cancers. Several studies have shown that higher NLR predicts a worse survival outcome in primary central nervous system lymphomas (PCNSLs) [1,2]. Previously, we also found that in a mixed cohort of primary and secondary central nervous system (CNS) lymphomas, higher pre-treatment NLR portended a worse overall survival (OS) [3]. This association, however, was not demonstrated in a Japanese cohort [4], which raised questions about the robustness of NLR across different patient populations. We, therefore, sought to investigate the use of NLR and other hematological indices in prognosticating survival for PCNSL in our population of patients.

Unlike other non-hematological malignancies, the processes that underlie changes in NLR may be due to share oncobiology for the lymphoma cells and the normal circulating lymphocytes. This is particularly of interest as the corticosteroid and chemotherapy agents used in the treatment of PCNSL directly alter the NLR and other hematological indices. Specifically, corticosteroid has been shown to cause profound lymphopenia, which may be due to the lympholytic effect on lymphoma cells. Corticosteroids are commonly used in reducing cerebral edema and form an important part of the treatment regime [5,6]. Corticosteroid-induced neutrophilia is also a well-described phenomenon, thought to be due to the demarginalization of neutrophils from the endothelial lining, reduced extravascular migration of neutrophils as well as an increased rate of release of immature neutrophils from the bone marrow into the circulation [7,8,9]. PCNSL is exquisitely sensitive to corticosteroids and has been termed ‘ghost tumor’ due to its tendency to disappear on neuroimaging following even a short duration of corticosteroids [10,11], which reduces diagnostic yield and confounds histological, immunohistochemical and cytological interpretation [12]. As such, steroids are typically withheld where possible until brain biopsy is successfully performed. Moreover, chemoimmunotherapy agents used in PCNSL can also alter the NLR and other hematological indices, and a better understanding of the changes in these indices during the course of treatment may enable a more refined and nuanced use of these indices. Lastly, as the effect of corticosteroids on circulating immune cell populations may mirror the response of lymphoma cells to corticosteroid treatment, hematological indices may also be a useful biomarker of response to subsequent chemotherapy.

In this study, we describe the prognostic value of post-steroid, pre-chemoimmunotherapy NLR and other hematological indices in PCNSL. We also trended the changes in these hematological indices before and after initiation of chemotherapy, and discussed their prognostic significance.

## 2. Methods

### 2.1. Patient Selection

Patients who underwent brain biopsy from January 2008 to December 2019 in two hospitals with tertiary neurosurgical service in Singapore were identified from surgery records and selected for the study. Patients aged 21 years and above with a histologically confirmed diagnosis of PCNSL were included. Patients who did not receive any chemoimmunotherapy, those with an unknown treatment regime (e.g., treated at an external center) or patients lost to follow-up, and those with no documented pre-chemoimmunotherapy steroid use were excluded. The final cohort consisted of patients who received chemotherapy or chemoradiotherapy and anti-CD20 (such as the combined MPV or R-MPV regimens, as described by Shah et al. and de Angelis et al. (“Shah protocol” and “de Angelis protocol”) and their variants [5,6], as well as newer regimens such as MATRIX [13]). The flowchart summarizing the patient selection is shown in Figure 1A. The typical treatment timeline is shown in Figure 1B.

All patients were followed up until 1 June, 2020 or death, whichever occurred first. This study was approved by the institutional review board (SingHealth IRB: 2018/3084).

### 2.2. Data Collection

Patient variables collected were: age, gender, Karnofsky performance status, Charlson Comorbidities Index (CCI) [14], the Memorial Sloan-Kettering Cancer Center (MSKCC) prognostic score [15] and the Nottingham/Barcelona score [16], Glasgow Coma Scale (GCS), location of lymphoma (categorized into lobar regions, corpus callosum, basal ganglia, thalamus, brainstem, and ependyma/meninges) and number of lesions (solitary versus multiple). Treatment-related variables included: pre-biopsy use of corticosteroids (yes or no), post-biopsy use of corticosteroids (yes or no, further stratified into high-dose ≥12 mg/day, or low-dose <12 mg/day), and chemotherapy regime (categorized into DeAngelis or Shah protocol or their variants, other methotrexate regimes, and others). Outcome variables included (from the date of biopsy): ECOG status at hospital discharge, 1 month, and 6 months, and date of death or last follow-up. For all Singapore residents, the mortality status and date of death were obtained from a state-maintained registry, the National Registry of Births and Deaths, via the National Records of Diseases maintained by the Immigration and Checkpoints Authority. For foreigners, the date of last follow-up was ascertained from the last available medical records.

### 2.3. Hematological Indices (HI)

For all patients, all available platelet counts, as well as absolute neutrophil (ANC), lymphocyte (ALC) and monocyte counts (AMC), were retrieved from electronic records from presentation (‘baseline’) until the date of last follow-up. The corresponding neutrophil-lymphocyte ratio (NLR), platelet-lymphocyte ratio (PLR), lymphocyte-monocyte ratio (LMR), and platelet-monocyte ratio (PMR) were calculated. For occasions where the monocyte count was 0, the values were set to 0.01 to prevent division by zero. Three time-points were used for stratification in the survival analysis: baseline (the earliest available blood test at the point of presentation), post steroids (steroids were routinely given immediately after adequate biopsy sample was obtained), and post chemoimmunotherapy (the blood test taken two weeks after initiation of treatment with chemoimmunotherapy).

### 2.4. Outcomes

Overall survival (OS), defined from the date of biopsy, was stratified by the respective HI, and estimated using the Kaplan–Meier method. The cut-off HI values were determined by the maximally selected log-rank statistics method with the corresponding cohort [17]. Log-rank tests were used to compare between the two Kaplan–Meier survival curves [18]. Patients were censored based on the time to last follow-up. Moreover, univariate as well as multivariate Cox regressions (adjusting for age and KPS, both components of the MSKCC score [15]) were performed [18]. Similarly, the time-to-progression (from the date of histological confirmation from biopsy) was stratified by the respective HI and estimated using the Kaplan–Meier method.

### 2.5. Trend and Correlation Analyses

Two types of analyses were then performed. Firstly, we investigated the trend of the hematological indices over time. The hematological ratios were pooled across the patient cohorts. All time-points for all patients were treated as a single cohort for such correlation analyses.

### 2.6. Statistical Analyses

Python version 3.7.1 was used to process data and perform statistical analyses. The lifelines library was used for survival analyses [19]. Median follow-up was estimated using the reverse Kaplan–Meier method [20]. For all analyses, the level of significance was set at *p* < 0.05 (two-tailed, prior to Bonferroni correction).

## 3. Results

### 3.1. Baseline Characteristics

We identified 119 patients with CNSL from surgical records. All patients received dexamethasone perioperatively. After excluding those who did not receive any form of chemoimmunotherapy (i.e., only received palliative WBRT or no treatment at all), lost to follow-up, and those with no documented pre-chemoimmunotherapy corticosteroid use, the final cohort for subsequent analyses consisted of 75 patients. The median OS was 30.3 months (95% CI: 25.5 to 54.8). The median PFS was 20.5 months (95% CI: 11.9 to 47.0). The median follow-up was 50.3 months (95% CI: 39.1 to 69.9).

Baseline demographics are summarized in Table 1. The median age of the patients in the final cohort was 61 years of age (range: 21 to 82), with a median KPS of 80 (range: 20 to 100). There was only one case of marginal zone lymphoma; the rest were all of the diffuse large B-cell lymphomas (DLBCL) subtype. Pre-biopsy steroid use was documented in 48% of individuals; 37% underwent methotrexate-based treatment with rituximab (“Shah regime”), 47% underwent methotrexate-based treatment without rituximab (“De Angelis regime”), and 16% received other chemoimmunotherapy regimes. The majority (69%) was classified as MSKCC class 2, 16% class 1 and 9% class 3.

### 3.2. Neutrophil-Lymphocyte Ratio (NLR)

For baseline NLR, taken prior to the initiation of corticosteroids and biopsy, the cut-off value of 2.0 was determined via the maximal log-rank statistics method [17]. There was no statistically significant difference in OS (log-rank *p*: 0.24, univariate Cox *p*: 0.25, multivariate Cox *p*: 0.19), although long-term survival appears to be better for those with NLR <2.0; however, the numbers with long-term data were small. For those with NLR <2.0, median OS was not reached by the end of 10 years (95% CI lower bound: 7.2 months), compared to those with NLR ≥ 2.0 (median OS 28.4 months, 95% CI: 17.9 to 47.0; log-rank *p*: 0.24) (Figure 2). Baseline NLR ≥ 2.0 was associated with a hazard ratio (HR) for a mortality of 1.69 (95% CI: 0.70–4.08, *p*: 0.25) in univariate Cox regression, and an adjusted HR of 1.85 (95% CI: 0.74–4.59, *p*: 0.19) in multivariate regression.

Pre-chemoimmunotherapy (post-steroid) NLRs were dichotomized with a higher threshold of 4.0, as the NLR values were generally higher following corticosteroids. Interestingly, there was a reversal in survival trend, with *higher* NLR being now associated with better median OS (47.0 months, 95% CI: 28.2 to not reached) as compared to lower NLR (26.5 months, 95% CI: 7.2 to 40.7 months; log-rank *p*: 0.03) (Figure 2). NLR ≥ 4.0 was associated with a hazard ratio (HR) for a mortality of 0.48 (95% CI: 0.25–0.95, *p*: 0.03) in univariate Cox regression, and an adjusted HR of 0.42 (95% CI: 0.21–0.83, *p*: 0.01) in multivariate regression.

NLR measured up to 2 weeks post-chemoimmunotherapy initiation dichotomized at an even higher value of 17.0 showed no statistically significant difference in OS (*p*: 0.42 in multivariate Cox regression), although those with NLR < 17.0 appeared to have a better OS than those with NLR > 17.0 at least in the short-to-medium terms (Figure 2).

In terms of tumor progression, a higher pre-chemoimmunotherapy (post-steroid) NLR (≥4.0) was associated with a borderline lower progression rate (median time to progression not reached by 10 years, 95% CI lower bound: 22.4 months), as compared to NLR <4.0 (median time to progression 20.5 months, 95% CI: 11.6 to 37.5 months; log-rank *p* = 0.07, multivariate-adjusted *p* = 0.18). We also performed analyses on patients before exclusion (i.e., all primary CNSL and those who received chemoimmunotherapy without steroids) and the results are in Appendix A. Of note, baseline NLR and post-treatment NLR were not found to be prognostic for tumor progression (Appendix A).

### 3.3. Platelet-Lymphocyte Ratio (PLR), Lymphocyte-Monocyte Ratio (LMR) and Platelet-Monocyte Ratio (PMR)

At baseline, none of the above indices showed OS differences (Figure 3).

Post steroids but pre chemoimmunotherapy, only lower LMR showed a statistically significant difference in OS, with LMR <25.7 having a better OS (median OS of 47.0 months, 95% CI: 25.5 to not reached) than those with LMR ≥25.7 (median OS of 27.5 months, 95% CI 2.9 to 40.7; *p*: 0.04) (Figure 3B). Univariate Cox HR was 1.97 for death (95% CI: 1.00 to 3.84; *p*: 0.05), and multivariate-adjusted HR was 2.17 (95% CI 1.10 to 4.31; *p*: 0.03). There was no OS difference for stratified PLR and PMR values (Figure 3B).

Post chemoimmunotherapy, only PLR, dichotomized at the threshold of 140, was associated with a borderline longer median OS (47.0 months, 95% CI: 25.5 to not reached) as compared to lower PLR (28.2 months, 95% CI: 10.3 to 40.7 months; log-rank *p*: 0.06) (Figure 3A). In Cox regressions, PLR ≥ 140 was associated with an univariate hazard ratio (HR) for a mortality of 0.52 (95% CI: 0.26 to 1.04, *p*: 0.06), and a multivariate-adjusted HR of 0.52 (95% CI: 0.26 to 1.04, *p*: 0.07).

### 3.4. Trend of Differential Counts and Hematological Indices (HI) over Time

Chemotherapy was started on a median of 11 days (IQR: 9 to 17 days) from the date of biopsy. The exact date of steroid administration was not reliably documented in the electronic records, especially for the earlier cohorts with no electronic medication charts, but it was routine in our institution to start (or continue, if preoperative steroids had already been given) post-surgical corticosteroids once a biopsy sample had been obtained. As such, we estimate that corticosteroids are started in a range of at least 9–17 days before chemoimmunotherapy initiation. The trend of changes in differential counts and HIs were shown in Figure 4. The absolute neutrophil count (ANC) increased up to the point of chemoimmunotherapy initiation before rapidly decreasing thereafter. The absolute lymphocyte count (ALC) decreased leading up to the chemoimmunotherapy initiation, plunged following chemoimmunotherapy induction, and remained low thereafter.

The absolute monocyte and platelet counts were relatively unchanged leading up to chemoimmunotherapy. After chemoimmunotherapy, the absolute monocyte count increased, with little change in the platelet count.

Owing to the known effects of neutrophilia and lymphopenia following steroid administration (Figure 4), there was a sharp increase in the NLR leading up to chemoimmunotherapy. A more gradual increase in PLR was also observed, driven mainly by the lymphopenia component, as platelet count did not change significantly prior to chemoimmunotherapy (Figure 4).

### 3.5. Comparison of the Change in Absolute Blood Differential Count and Hematological Indices Pre and Post Steroid

Further comparison of the change in absolute differential blood count revealed that only the change in monocyte count after steroids correlated significantly with OS, with a post-steroid-to-pre-steroid monocyte ratio of greater than 0.2 correlating with better OS (Appendix A). Furthermore, while not statistically significant, a post-steroid- -to-pre-steroid lymphocyte ratio of <0.8 was associated with worse OS, while a ratio of ≥0.6 in neutrophils was associated with better OS (Appendix A).

In terms of HI, only the relative change in NLR pre and post steroid reached borderline statistical significance, with a larger increase in NLR (≥2.5) correlating with better OS (age- and KPS-adjusted Cox regression *p* = 0.05) (Appendix A). None of the other HIs were prognostic for OS. Taking into account the relative change in absolute lymphocyte and neutrophil counts pre and post steroids, this pointed towards the possibility that the relative *balance* of lymphocyte and neutrophil composition was more prognostic than either component alone.

### 3.6. Comparison of the Change in Absolute Blood Differential Count and Hematological Indices Pre and Post Chemoimmunotherapy

Comparing post-chemoimmunotherapy blood counts with pre-chemoimmunotherapy counts, only the change in relative monocyte count correlated with OS, with a lower fall in absolute monocyte count (≥0.2 post versus pre chemoimmunotherapy) correlating with better OS (log-rank *p* = 0.02, multivariate-adjusted *p* = 0.03) (Appendix A). For HI, none of the relative changes in hematological indices correlated with OS differences (Appendix A).

### 3.7. The National Cancer Center Singapore—National Neuroscience Institute (NCCS-NNI) Prognostic Model

Based on our findings, we designed a prognostic model to predict two-year OS. A decision tree model was trained using a 60:40 train–test sample split for patients who subsequently received chemotherapy as well as those who received interim corticosteroids prior to chemotherapy (*n* = 75). Those with missing data were excluded, and a final sample of 56 patients with complete data was available for model training and testing (train set *n* = 33, test set *n* = 23). A decision tree model was used to generate a predictive score, building on the previously described MSKCC score [15]. As the median OS was roughly 2 years, 2-year mortality was chosen as the outcome variable for decision tree model training. As the baseline (pre-steroid) NLR, pre-chemoimmunotherapy (post-steroid) NLR and LMR were shown to be good risk-stratifiers in the published literature [1,2] and our own survival analysis, they were added to the age and KPS (or ECOG) parameters common to both the MSKCC and the Nottingham/Barcelona scores as candidate variables.

A new prognostic score, the NCCS-NNI score, was thus proposed (Figure 5), incorporating age and the newly identified variables: pre- and post-steroid (both pre-chemoimmunotherapy) NLR. KPS and LMR were eliminated by the decision tree model during the training process. When internally validated on a hold-out test set not used in the decision tree training (*n* = 23), the decision tree showed good discrimination with an accuracy of 0.78 and area-under-curve (AUC) of 0.78 (Figure 5B). The model was well calibrated with a Brier score of 0.163 (Figure 5C). For the NCCS-NNI score, the low-risk class (class 1) corresponded to a 2-year mortality of 5%, the medium-risk class (class 2) to 38% and the high-risk class (class 3) to 73%. Kaplan–Meier analysis over a 10-year period (noting that the decision tree model was only trained on 2-year mortality) showed that the risk stratification persisted, especially between class 1 and 3 (*p* < 0.005 for log-rank test over the 10-year curves) and class 1 and 2 (*p* = 0.01), but less so for class 2 and 3 (*p* = 0.11) as OS converged to about 20% beyond 4 years. Furthermore, in our cohort, the MSKCC and the Nottingham/Barcelona scores poorly correlated with 2-year mortality, and none of the survival trends estimated by Kaplan–Meier analysis were significantly different for either score (Figure 6). The AUC for MSKCC was 0.500 and that for the Nottingham/Barcelona score was 0.424 (Appendix A). The concordance index (comparing the respective median survivals) for the NCCS-NNI against the MSKCC scores was 0.447, and that against the Nottingham/Barcelona score was 0.688.

## 4. Discussion

As HIs are increasingly being used as prognostic markers in many cancers, we have sought to characterize the prognostic value of HIs in PCNSL as well as changes in the underlying blood cell population at significant timepoints during the patient journey. In addition, PCNSL is a unique entity, as corticosteroids (dexamethasone in most cases) are often started empirically for reducing cerebral edema. As we have shown in this bi-institution study with a relatively large sample size for an uncommon disease entity such as PCNSL, the well-known lymphotoxic properties of corticosteroids have lent themselves serendipitously to aid in survival prognostication. We found that higher NLR following steroids (as compared to baseline) correlated with better OS (Appendix A, multivariate Cox *p*: 0.05). In fact, there is a reversal of effect direction of the post-steroid NLR—with higher NLR after corticosteroids correlating with better OS as well as lower tumor progression rate—as compared to the baseline NLR, the latter of which echoed previous results where higher NLR correlated with worse OS. Finally, a new prognostic score (NCCS-NNI score) incorporating the pre- and post-steroid NLR and age demonstrated good discriminative power and was well calibrated in our internal validation.

### 4.1. Relationship of NLR and Survival

In a previous, more general cohort of primary and secondary CNS lymphomas, comprised of both patients who were treated with chemo-radiotherapy and those who received palliative radiotherapy only, we found that higher NLR portended a worse OS [3]. Here, we demonstrated the same trend in a pure PCNSL cohort. This was in keeping with several studies by other author groups [1,2,21]. In the Jung et al. study [1], NLR greater or equal to 2.0 was associated with worse survival; of note, we also arrived at the same cut-off value of 2.0 for our baseline NLR, with effect estimates in the same direction. Similarly, in a more recent Luo et al. study [21], the group found that NLR > 1.79 correlated with worse PFS.

Other than the NLR obtained at initial presentation, a natural timepoint to prognosticate would be just prior to the commencement of chemotherapy, where patients would typically have been started on steroids post biopsy. In our cohort, there was a short interval—a median of 11 days—between biopsy and chemotherapy commencement. Paradoxically, we observed a trend reversal in NLR after the patients were started on corticosteroids, with a high post-steroid NLR actually favoring a *better* prognosis (and a lower tumor progression rate, albeit only reaching borderline statistical significance). Analysis of the underlying blood cell differentials revealed that both the fall in ALC as well an increase in ANC drove this reversal in NLR. Interestingly, a larger fall in ALC post versus pre steroids (namely with a ratio of ≤0.8) was non-significantly associated with *worse* OS, while a larger post- versus pre-steroid ANC ratio (of ≥0.6) was non-significantly associated with *better* OS (Appendix A). This was consistent with the known, and in fact commonly observed, hematological effects of corticosteroids, which we will further elaborate in the next section.

### 4.2. The Effect of Corticosteroids on Blood Cells

The neutrophilic and the direct lympholytic properties of corticosteroids have been well described [7,22]. Corticosteroids are potent inducer of apoptosis in lymphoma cells, mediated by the regulation of apoptosis genes (e.g., members of the Bcl-2 family), and their binding to cytoplasmic steroid receptor and their subsequent translocation to the nucleus to regulate gene expression related to cell death or survival [23]. The potent lympholytic property of corticosteroids has consequently been used in the treatment of lymphoid malignancies [24]. In extreme cases, prolonged remission of PCNSL even after a short duration of corticosteroids has been reported [25,26]. Time trend analysis revealed that the elevated NLR pre-chemoimmunotherapy was driven by both a decrease in ALC and an increase in the ANC (Figure 4), where most patients received corticosteroids (only four of the 79 patients were not known to receive any steroids, and similar results were obtained after these four patients were excluded). The combined neutrophilia and lymphopenia caused an increase in NLR after corticosteroid administration (Figure 4). From our survival analysis, we show that patients who experienced such an increase in NLR tended to have better survival, as well as lower tumor progression rate—the same cellular mechanisms that led to the observed systemic lymphopenia after corticosteroids likely also induced apoptosis of PCNSL cells, and response to steroids could serve as a proxy for tumor sensitivity to subsequent chemotherapeutic agents.

The role of neutrophils in the context of PCNSL is less clear. There was some evidence that the efficacy of rituximab is dependent on neutrophils, as seen in the blunted response to the anti-CD20 antibody rituximab, in the neutrophil-depleted B-cell lymphoma mouse model [27]. One hypothesis could be that neutrophils at baseline are pro-tumorigenic, which explains why baseline high NLR is poorly prognostic, and why the well-established efficacy of rituximab is blunted in a neutrophil-depleted model. However, neutrophils induced by steroids are not necessarily pro-tumorigenic [7] and hence are distinct from tumor-associated neutrophils.

### 4.3. NCCS-NNI Prognostic Model

We found that the MSKCC score was poorly discriminatory in our population. One key difference between our cohort and that used to derive the MSKCC was that the latter was developed based on patients enrolled in the 1980s to early 2000s, prior to the advent of the modern Shah and De Angelis regimes [15], as well as the use of rituximab [6,28]. Modern chemoradiotherapy conferred a significantly better survival than the chemotherapy available in the MSKCC cohort. Furthermore, in our population, patients tended to present relatively early in their clinical course with easy early access to a specialist health service [3], with relatively good baseline performance status (median of 80, compared to 70 in the MSKCC cohort). As shown in our previous work [3], our cohort also generally had a slightly larger proportion of patients who presented with focal motor deficits (34%, versus 28% who presented with hemiparesis in the MSKCC cohort).

Given these differences in patient characteristics, MSKCC score was only weakly applicable in our population. The poor calibration of MSKCC to modern patient cohorts was also noted by Luo et al. in a Chinese cohort from the rituximab era, associated with a poor concordance index of 0.57 [21]. Our new proposed NCCS-NNI score was more discriminative and better calibrated in our internal validation, and could be more in line with modern trends in healthcare and treatment options.

A decision tree model took into account the non-linear interactions between the variables. For instance, age above 70 alone places an individual into the high-risk group, regardless of the hematological variables. For patients under 70, the absence of a raised NLR post steroid (being a composite for relative lymphopenia and neutrophilia) placed the individual in the medium-risk group. The low-risk group was populated by young individuals (less than 70 years old), with an elevated NLR (>4.0) following steroids, but a low NLR at baseline (<6.9). The KPS used in the MSKCC score was not considered by the decision tree model to be a discriminatory variable.

### 4.4. Strengths and Limitations

A near-complete follow-up was achieved as mortality data were retrieved from the state registry. On the proposed NCCS-NNI score, the discriminatory power was good, and when tested on a separate test set of patients not used in model training, it was also well calibrated (Brier score of 0.16).

The exact timing of dexamethasone initiation could not be determined, as the archived prescription charts were not available. The OS trend of the medium- (class 2) and high-risk groups (class 3) also converged around the 4-year mark, indicating that these risk strata did not persist beyond the 2-year period that the decision tree model was trained and internally validated on. Nevertheless, low-risk (class 1) patients continued to have good OS up to 10 years post diagnosis.

As for any predictive model, this model would only apply to similar populations. In our opinion, the model would be most suitable for patients likely to undergo chemoradiotherapy, and possibly when patients present relatively early in their disease course (as was our population). With new treatment options and a better understanding of PCNSL as a disease since the development of scores such as the MSKCC and the Nottingham/Barcelona scores, perhaps our centers were able to better select patients to undergo biopsy. Those deemed too frail to undergo subsequent treatment, even if proven to be PCNSL, might not be recommended to undergo a biopsy in the first place. The observation that few high-risk patients (as determined by the MSKCC and the Nottingham/Barcelona scores) had been included might be suggestive of this. There is likely to be varied characteristics of patients observed at different centers. As such, it could be challenging to develop prognostic models that are globally applicable. Nevertheless, the value of NLR (specifically pre chemoimmunotherapy and post steroid) is demonstrated in our patient populations across two separate centers. Unlike other factors in established models, the NLR could serve as a potential prognostic factor that is dynamic in its response to steroids.

Lastly, more sophisticated machine learning models, such as random forest, support vector machine or neural net, and/or the inclusion of more variables, could in principle give a better prediction, but these models tend to be “black box” and cumbersome to use in routine clinical practice; a decision tree model, therefore, preserves transparency and fits the clinical heuristics, at the expense of poorer predictive performance.

## 5. Conclusions

We have performed a bi-institutional study on a relatively large cohort of PCNSL, a rare entity. NLR is an important prognostic variable to consider. An increase in NLR after steroid administration could be a biomarker for favorable survival and a lower progression rate after the initiation of chemotherapy. The NCCS-NNI prognostic model outperformed the MSKCC and the Nottingham/Barcelona score in our cohort. It would be valuable to evaluate its use in other patient populations.

## Figures and Tables

**Figure 1 cancers-14-01818-f001:**
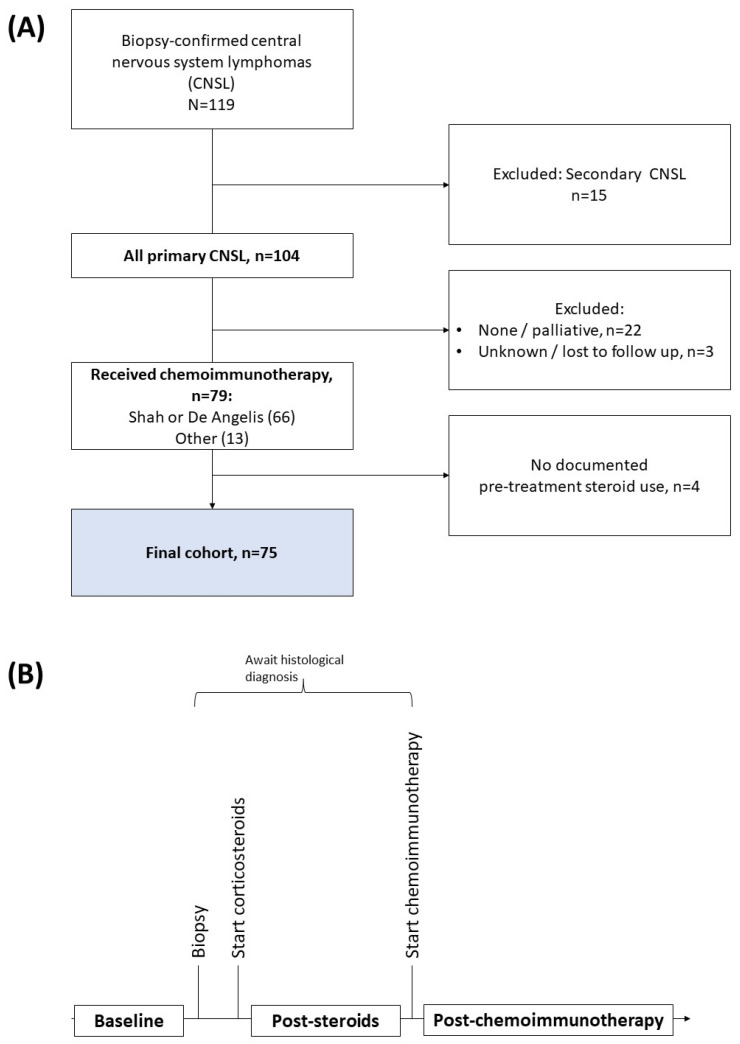
(**A**) Flowchart of patient inclusion and exclusion. Kaplan–Meier survival analyses were performed on the final cohort. (**B**) Timeline of the typical treatment algorithm of primary CNSL.

**Figure 2 cancers-14-01818-f002:**
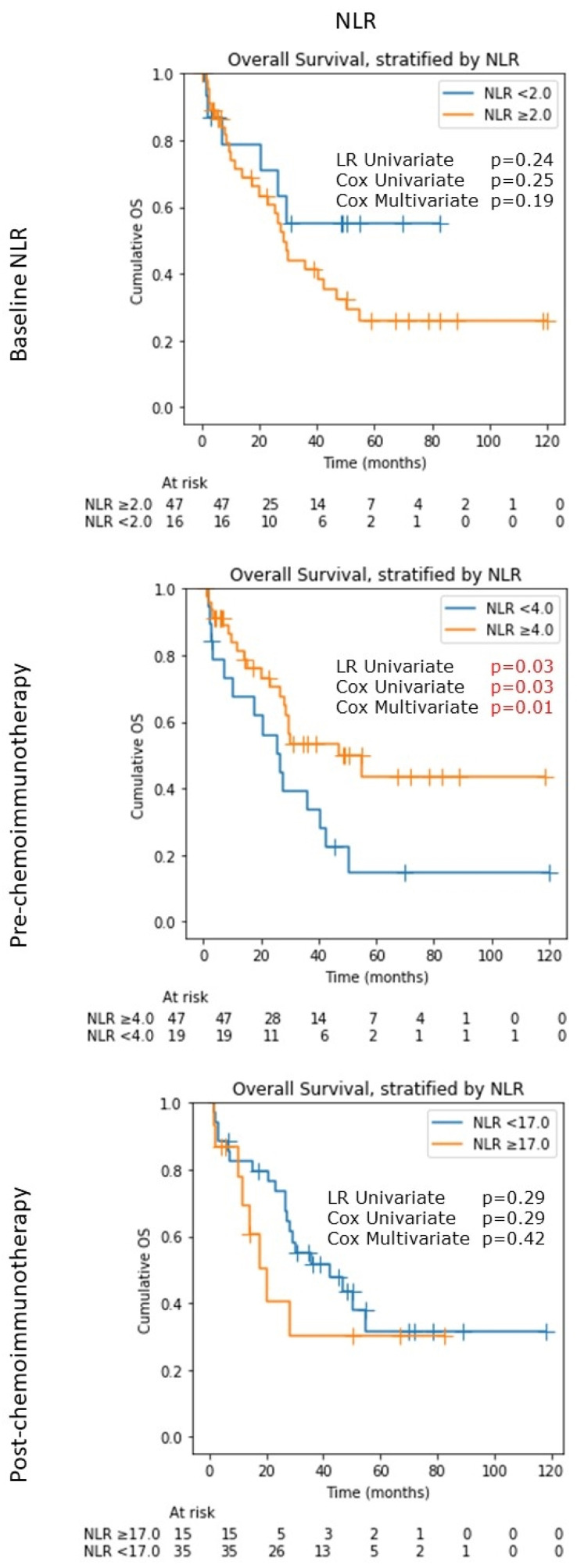
Kaplan–Meier (KM) plot for overall survival stratified by NLR values at three time-points: baseline pre steroids, pre chemotherapy post steroids, and post chemotherapy. For each KM plot, three statistical tests were performed: log-ranked test (LR) on the two survival curves based on the corresponding optimal cut-off value, univariate Cox regression, and multivariate Cox regression (adjusted for age and Karnofsky performance status) at the same cut-off value. NLR: neutrophil-lymphocyte ratio.

**Figure 3 cancers-14-01818-f003:**
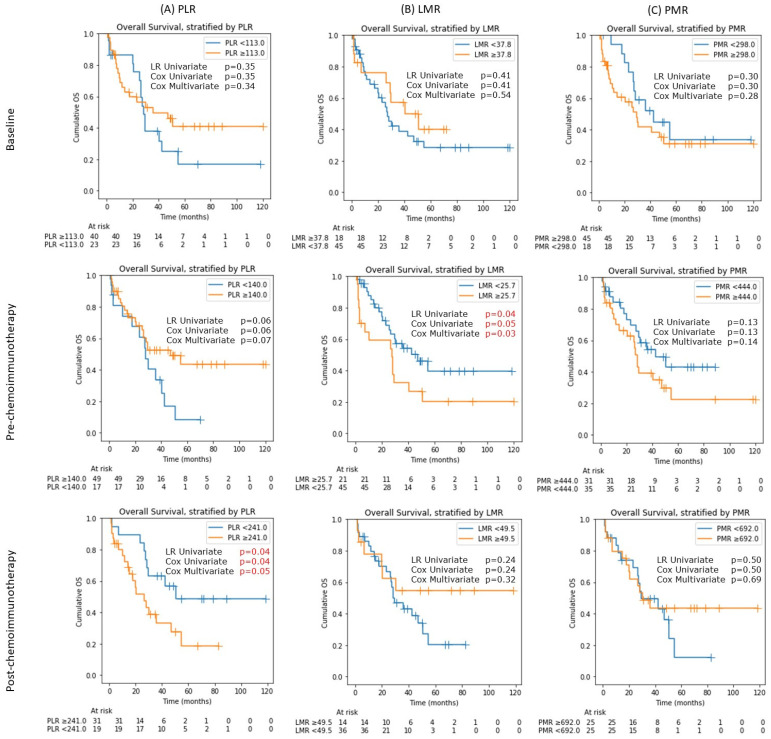
Kaplan–Meier (KM) plot for overall survival stratified by PLR, LMR and PMR values at three time-points: baseline pre steroids, pre chemotherapy post steroids, and post chemotherapy. For each KM plot, three statistical tests were performed: log-ranked test (LR) on the two survival curves based on the corresponding optimal cut-off value, univariate Cox regression, and multivariate Cox regression (adjusted for age and Karnofsky performance status) at the same cut-off value. PLR: platelet-lymphocyte ratio; LMR: lymphocyte-monocyte ratio; and PMR: platelet-monocyte ratio.

**Figure 4 cancers-14-01818-f004:**
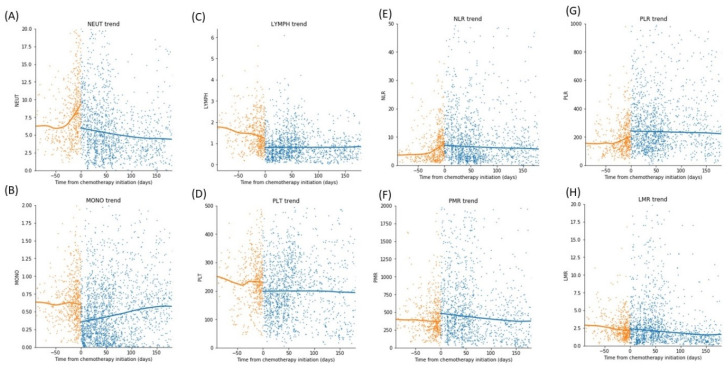
The time trend of differential counts of blood cells over time, separately fitted with locally weighted scatterplot smoothing (LOWESS) curve before and after chemotherapy initiation, of (**A**) absolute neutrophil count, (**B**) absolute monocyte count, (**C**) absolute lymphocyte count, (**D**) platelet count, (**E**) neutrophil-lymphocyte ratio, (**F**) platelet-monocyte ratio, (**G**) platelet-lymphocyte ratio, and (**H**) lymphocyte-monocyte ratio. All differential counts were expressed in units of 10^9^ cells per liter. The absolute neutrophil count (NEUT) increased up to the point of chemotherapy initiation before rapidly decreasing thereafter, and the absolute lymphocyte count (LYMPH) decreased leading up to the chemotherapy initiation. The absolute monocyte (MONO) and platelet (PLT) counts were relatively unchanged leading up to the chemotherapy. The neutrophil-lymphocyte ratio (NLR) increased sharply leading up chemotherapy initiation before slowly declining thereafter. The platelet-lymphocyte ratio (PLR) also increased but at a slower rate, whereas the platelet-monocyte ratio (PMR) and lymphocyte-monocyte ratio (LMR) remained largely unchanged leading up to the chemotherapy.

**Figure 5 cancers-14-01818-f005:**
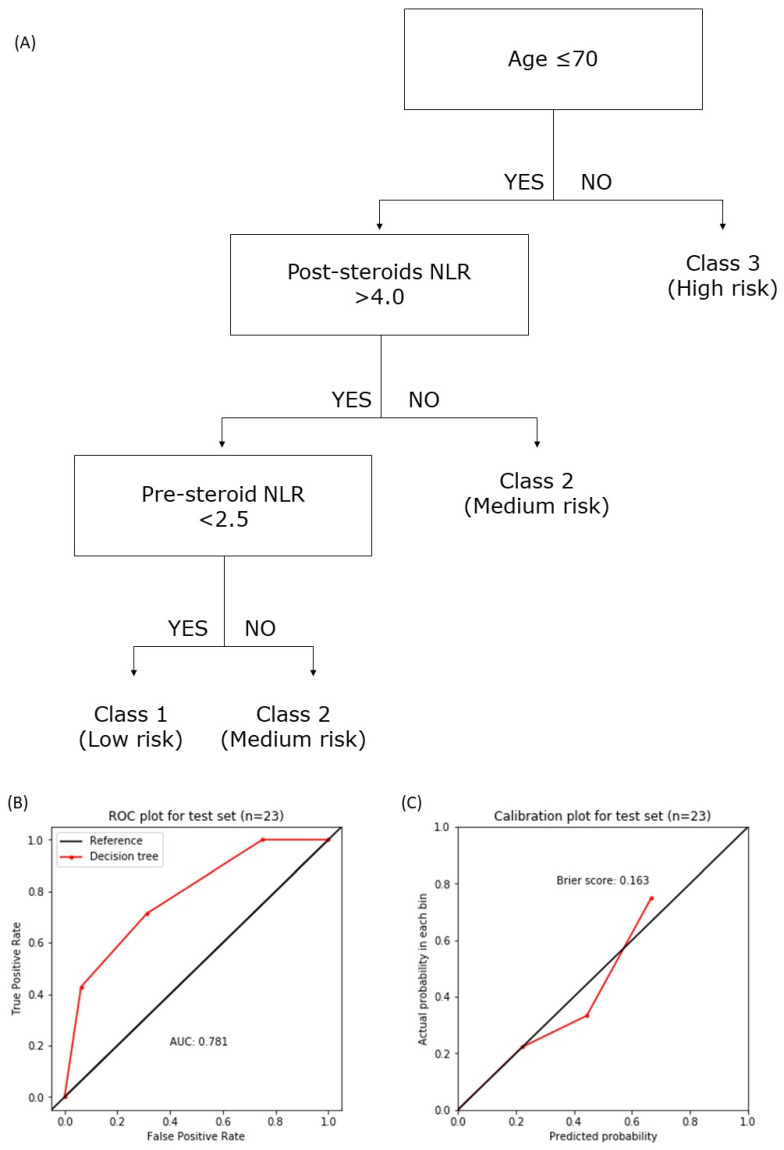
Decision tree model. (**A**) A simple decision-tree model trained and tested on a 60–40 train–test split of the data. The algorithm classifies the patients into three risk classes: low- (class 1), medium- (class 2) and high-risk (class 3) patients. Five candidate features were included in training: age, KPS, baseline NLR, post-steroid/pre-chemotherapy NLR, and the post-steroid change in absolute lymphocyte count. The KPS and baseline NLR were not identified by the decision tree model to be an informative predictor in our cohort and were hence not included. The cut-off points were determined by the decision tree model. KPS: Karnofsky Performance Status; NLR: neutrophil-lymphocyte ratio. (**B**) Decision tree receiver operating characteristic (ROC) curve, with an area-under-curve (AUC) of 0.840 and (**C**) calibration plot, with a Brier score of 0.149, internally validated on the test set.

**Figure 6 cancers-14-01818-f006:**
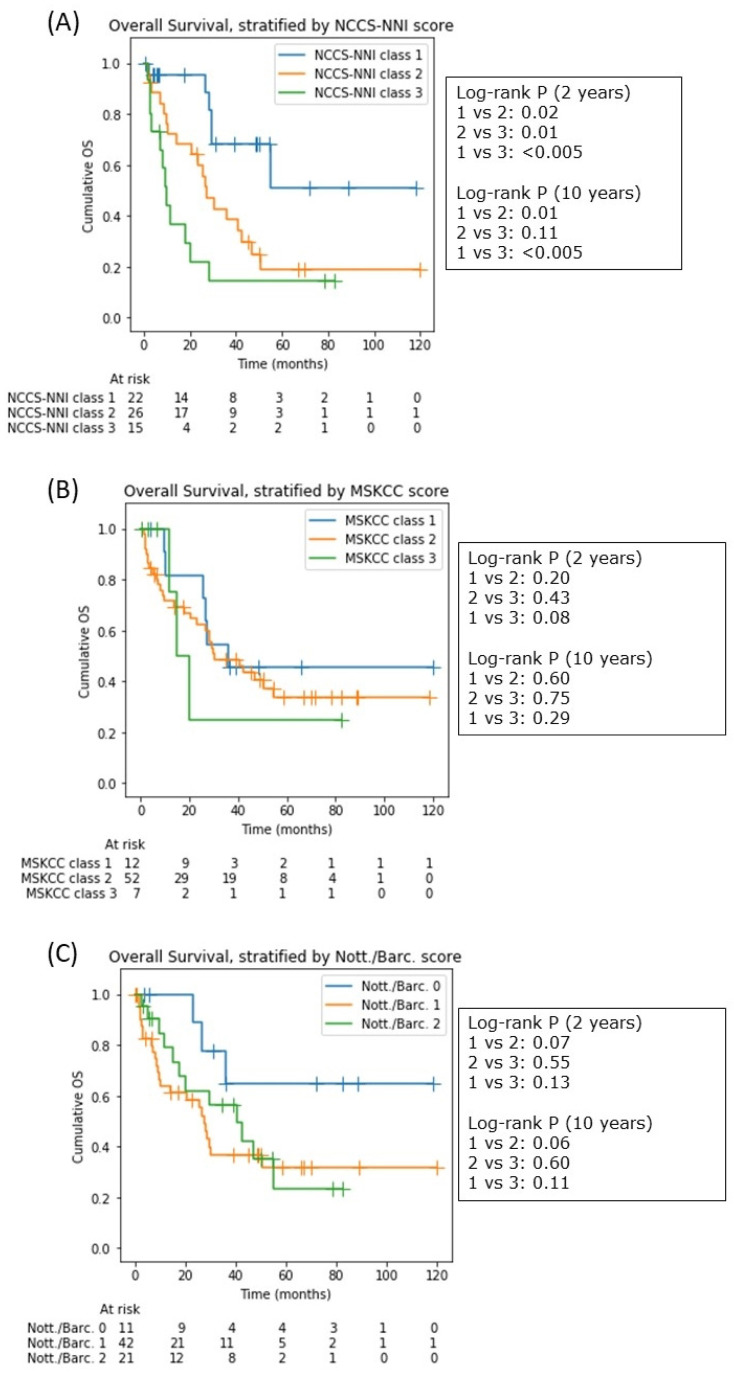
Kaplan–Meier survival analysis based on the (**A**) NCCS-NNI score, (**B**) the MSKCC score, and (**C**) the Nottingham/Barcelona score. The difference in OS, as stratified by the NCCS-NNI score persisted into 10 years post diagnosis, although for NCCS-NNI class 2 and 3 the survival curves converged at approximately 4 years. Note that there were no patients with a Nottingham/Barcelona score of 3.

**Table 1 cancers-14-01818-t001:** Baseline demographics of the patient cohort. CCI: Charlson Comorbidities Index; IQR: Interquartile range; KPS: Karnofsky Performance Status; MSKCC: Memorial Sloan-Kettering Cancer Center. MTX: Methotrexate.

Parameters	Final Cohort (*n* = 75)
Age, median (range)	61 (21–82)
Male gender, *n* (%)	41 (55)
KPS, median (range)	80 (20–100)
CCI, median (range)	2 (0–6)
MSKCC prognostic score, *n* (%)	
Class 1	12 (16)
Class 2	52 (69)
Class 3	7 (9)
Pre-biopsy dexamethasone use, *n* (%)	36 (48)
Post-biopsy dexamethasone use, *n* (%)	
None	4 (5)
Low-dose (<12 mg/day) or delayed	8 (11)
High-dose (≥12 mg/day)	63 (84)
Chemoimmunotherapy regime, *n* (%)	
Shah (MTX with Rituximab)	28 (37)
De Angelis (MTX without Rituximab)	35 (47)
Other (e.g., MATRIX)	12 (16)

## Data Availability

The data could not be made publicly available as per our research policies.

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
