# Peer review of "A Prognostic Model Using Post-Steroid Neutrophil-Lymphocyte Ratio Predicts Overall Survival in Primary Central Nervous System Lymphoma"

_cancers, 2022, doi:10.3390/cancers14071818_

Round 1
Reviewer 1 Report
The authors evaluate the prognostic value of several hematological indices in a series of 75 primary central nervous system lymphomas, at different points of the evolution of their treatment. They found that post-steroid, pre-immunochemotherapy, neutrophil-lymphocyte ratio (NLR) has a prognostic value. Moreover, they propose a new prognostic score based on age and NLR pre and post-steroid (pre-immunochemotherapy) treatment.
The study is well done, the results are interesting, and the manuscript is clear. My comments / recommendations are very minor:
- Regarding patients selection, the authors state that they include patients with a histologically confirmed diagnosis of primary central nervous system lymphoma (PCNSL). Although most lymphoma occurring exclusively in the CNS are of diffuse large B-cell lymphoma type, a minority of CNS-based lymphomas are MALT lymphoma or “double-hit” lymphomas with MYC and BL2 and/or BCL6 rearrangements or others. The authors should confirm at some point if all cases corresponded to diffuse large B-cell lymphoma type.
- Line 233-4: the authors state that “it was routine in our institution to start post-surgical corticosteroid once a biopsy sample had been obtained”. This seems contradictory with line 159: “Pre-biopsy steroid use was documented in 48% of individuals”.
Author Response
Thank you for your helpful comments to improve our manuscript.
Our responses to your specific comments are as follow.
--
The authors evaluate the prognostic value of several hematological indices in a series of 75 primary central nervous system lymphomas, at different points of the evolution of their treatment. They found that post-steroid, pre-immunochemotherapy, neutrophil-lymphocyte ratio (NLR) has a prognostic value. Moreover, they propose a new prognostic score based on age and NLR pre and post-steroid (pre-immunochemotherapy) treatment.
The study is well done, the results are interesting, and the manuscript is clear. My comments / recommendations are very minor:
- Regarding patients selection, the authors state that they include patients with a histologically confirmed diagnosis of primary central nervous system lymphoma (PCNSL). Although most lymphoma occurring exclusively in the CNS are of diffuse large B-cell lymphoma type, a minority of CNS-based lymphomas are MALT lymphoma or "double-hit" lymphomas with MYC and BL2 and/or BCL6 rearrangements or others. The authors should confirm at some point if all cases corresponded to diffuse large B-cell lymphoma type.
RESPONSE:
Thank you for your valuable comment. Indeed, most PCNSLs were DLBCL except for one case of marginal zone lymphoma. There was no double-hit lymphomas with gene rearrangements that we identified.
We have included the above into our manuscript as follow (Lines 182-183):
“There was one case of marginal zone lymphoma; the rest were all of the diffuse large B-cell lymphomas (DLBCL) subtype.”
- Line 233-4: the authors state that "it was routine in our institution to start post-surgical corticosteroid once a biopsy sample had been obtained". This seems contradictory with line 159: "Pre-biopsy steroid use was documented in 48% of individuals".
RESPONSE:
This statement only pertained to the post-surgical use of steroids, which could be either initiation or continuation of preoperative steroids. Practically in many cases (i.e. the 48% of cases) preoperative steroids were started based on the findings of symptomatic vasogenic edema on the initial plain CT brain scan for relief of raised ICP symptoms. These patients were evaluated further with contrasted MRI brain which then raised the suspicion of PCNSL, for which steroids would be typically stopped. (Note that the exact circumstances could not be determined accurately due to the clinical records not being available especially for the older cohorts. This piece of information should not however change our conclusion).
We have clarified this by adding the following phrase for clarification (Lines 288-289): “It was routine in our institution to start (or continue, if preoperative steroids had already been given) post-surgical corticosteroid once a biopsy sample had been obtained.”
Reviewer 2 Report
The neutrophil-to-lymphocyte ratio (NLR), a biomarker for systematic inflammation, has been recently identified as a prognostic factor for various types of both solid and hematologic malignancies including NHLs.
Results of this study which enrolled 75 PCNSL patients who received chemoimmunotherapy suggest that NLR, assessed pre-chemoimmunotherapy (post-steroid), was predictive of OS. Furthermore, a decision-tree model including age ≤70, post-steroid NLR >4.0, and pre-steroid (baseline) NLR <2.5 allowed a separation into 3 different prognostic groups. Of note, the area-under-curve value denotes the good performance of the model.
Authors conclude that post-steroid NLR has prognostic in PCNSL and can be incorporated into a model useful to better stratify from a prognostic standpoint these patients whose management is still challenging.
MAJOR POINTS
The authors conclude that the NCCS-NNI prognostic model outperformed the MSKCC and the Nottingham/Barcelona score in their cohort. However, these conclusions should be better supported by a comparison performed using the Akaike information criteria (AIC).
I suggest performing also a concordance analysis between the model developed in this study and MSKCC or the Nottingham/Barcelona score in order to verify how patients with low, intermediate, and high risk are distributed in the different models.
The authors correctly discuss the limitations of their study. In my view the applicability of this score in other patient cohorts is limited. In the context of NCCS relevant are the differences in the characteristics of patients observed at different centers. This heterogeneity makes it really difficult to develop “globally applicable” prognostic models.
This point should be discussed in detail.
For the above-mentioned reasons I suggest reducing the emphasis on the prognostic model underscoring, in contrast, the value of NLR, assessed pre-chemoimmunotherapy (post-steroid), as a potential dynamic prognostic factor in NCCS
.
Author Response
Thank you for your helpful comments to improve our manuscript.
Our responses to your specific comments are as follow.
---
The neutrophil-to-lymphocyte ratio (NLR), a biomarker for systematic inflammation, has been recently identified as a prognostic factor for various types of both solid and hematologic malignancies including NHLs.
Results of this study which enrolled 75 PCNSL patients who received chemoimmunotherapy suggest that NLR, assessed pre-chemoimmunotherapy (post-steroid), was predictive of OS. Furthermore, a decision-tree model including age ≤70, post-steroid NLR >4.0, and pre-steroid (baseline) NLR <2.5 allowed a separation into 3 different prognostic groups. Of note, the area-under-curve value denotes the good performance of the model.
Authors conclude that post-steroid NLR has prognostic in PCNSL and can be incorporated into a model useful to better stratify from a prognostic standpoint these patients whose management is still challenging.
MAJOR POINTS
The authors conclude that the NCCS-NNI prognostic model outperformed the MSKCC and the Nottingham/Barcelona score in their cohort. However, these conclusions should be better supported by a comparison performed using the Akaike information criteria (AIC).
RESPONSE:
Thank you for your comment. AIC would not be suitable for decision-tree based models like ours or RPA like MSKCC. However for the same purpose, we have calculated the AUC for MSKCC and the Nottingham/Barcelona scores (0.500 and 0.424 respectively).
We have added this line in our Results section (Lines 372-373): “The AUC for MSKCC was 0.500 and that for the Nottingham/Barcelona score was 0.424 (Supplementary Figure 6).”
I suggest performing also a concordance analysis between the model developed in this study and MSKCC or the Nottingham/Barcelona score in order to verify how patients with low, intermediate, and high risk are distributed in the different models.
RESPONSE:
Thank you for your comment. Additional analyses using concordance analysis have now been performed.
We have added the results as follow (Lines 373-375): “The concordance index (comparing the respective median survivals) for the NCCS-NNI against the MSKCC scores was 0.447, and that against the Nottingham/Barcelona score was 0.688.”
The authors correctly discuss the limitations of their study. In my view the applicability of this score in other patient cohorts is limited. In the context of NCCS relevant are the differences in the characteristics of patients observed at different centers. This heterogeneity makes it really difficult to develop "globally applicable" prognostic models.
This point should be discussed in detail.
For the above-mentioned reasons I suggest reducing the emphasis on the prognostic model underscoring, in contrast, the value of NLR, assessed pre-chemoimmunotherapy (post-steroid), as a potential dynamic prognostic factor in NCCS
RESPONSE:
Thank you for your comment. We agree and we have now included this point in our discussion (Lines 521-527): "There are likely to be varied characteristics of patients observed at different centers. As such, it could be challenging to develop prognostic models that are globally applicable. Nevertheless, the value of NLR (specifically pre-chemoimmunotherapy and post-steroid) is demonstrated in our patient population across two separate centers. Unlike other factors in established models, the NLR could serve as a potential prognostic factor that is dynamic in its response to steroids.”
Reviewer 3 Report
REVIEW TO THE ORIGINAL PAPER:
A prognostic model using post-steroid neutrophil-lymphocyte ratio predicts overall survival in primary central nervous system lymphoma; authors: Lo et al.
The presented original paper is well written, good structured, with interesting focus and has a scientific as well as practical impact. Generally, I recommend warmly this article for publication. I have only couple of comments.
Major:
Please could the authors add median follow-up of the whole cohort of PCNSL patients, global ORR, OS and PFS for all patients? Thanks
Minor:
- Methods/2.2. data collection (line 110) „….high-dose corticosteroids ≥12mg/day, or low-dose <12mg/day …“ I recommend to add what corticosteroids are meant – probably Dexamethasone….?
- Methods/2.4. Outcomes (line 130) „….Overall survival (OS), defined from the date of histological confirmation from biopsy, 130 was stratified by…“ I prefer to caculate the data of initila diagnosis from the date of biopsy (performed) and not from the date from the confirmation
- Results/3.1. Baseline characteristics (lines 150-156) I recommend to omit or substantially reduce this part of text, all the data are described well in the Figure 1. This information is redundant (duplicated).
- Baseline demographic (Table 1), please add gender and ethnicity structure. Further, in some categories the „%“ is missing, and again I recommend to put to the dosage of cortisterois exact type eg. Dexamethasone)
- Generally, for the descriptive values, there is the best with median to present ranges (min-max), quartilles are possible, but very difficult for comparison with other published data
Author Response
Thank you for your helpful comments to improve our manuscript.
Our responses to your specific comments are as follow.
---
A prognostic model using post-steroid neutrophil-lymphocyte ratio predicts overall survival in primary central nervous system lymphoma; authors: Lo et al.
The presented original paper is well written, good structured, with interesting focus and has a scientific as well as practical impact. Generally, I recommend warmly this article for publication. I have only couple of comments.
Major:
Please could the authors add median follow-up of the whole cohort of PCNSL patients, global ORR, OS and PFS for all patients? Thanks
RESPONSE:
Thank you for your comment. These survival figures have been added as follows (except for ORR as this database was not designed to capture response rate):
Methods, Lines 168-170: “Median follow-up was estimated using the reverse Kaplan-Meier method [20].”
Results, Lines 178-180: “The median OS was 30.3 months (95% CI: 25.5 to 54.8). The median PFS was 20.5 months (95% CI: 11.9 to 47.0). The median follow-up was 50.3 months (95% CI: 39.1 to 69.9).”
Minor:
Methods/2.2. data collection (line 110) "….high-dose corticosteroids ≥12mg/day, or low-dose <12mg/day …" I recommend to add what corticosteroids are meant – probably Dexamethasone….?
RESPONSE:
Thank you for your comment. All patients were given dexamethasone perioperatively. Some of them could have been converted to other corticosteroids such as prednisolone later on in the chemoradiotherapy treatment, but this data was not captured in our database and would not affect our conclusions as these timepoints were beyond those we used to calculate the hematological indices.
We have added the following into the Results section (Lines 174-175) to clarify this: “All patients received dexamethasone perioperatively”.
Methods/2.4. Outcomes (line 130) "….Overall survival (OS), defined from the date of histological confirmation from biopsy, 130 was stratified by…" I prefer to caculate the data of initila diagnosis from the date of biopsy (performed) and not from the date from the confirmation
RESPONSE:
Thank you for your comment. The dates were indeed calculated from the biopsy date. The wording had been clarified as such “...defined from the date of biopsy...” (Line 153), removing the word “histological confirmation” (it was implied that biopsy led to the eventual histological confirmation).
Results/3.1. Baseline characteristics (lines 150-156) I recommend to omit or substantially reduce this part of text, all the data are described well in the Figure 1. This information is redundant (duplicated).
RESPONSE:
Thank you for your comment. We have shortened the paragraph to the following (Lines 174-180), with some additional information requested by the other reviewers): “We identified 119 patients with CNSL from surgical records. All patients received dexamethasone perioperatively. After excluding those who did not receive any form of chemoimmunotherapy (i.e. only received palliative WBRT or no treatment at all), lost to follow up, and those with no documented pre-chemoimmunotherapy corticosteroids use, the final cohort for subsequent analyses consisted of 75 patients. The median OS was 30.3 months (95% CI: 25.5 to 54.8). The median PFS was 20.5 months (95% CI: 11.9 to 47.0). The median follow-up was 50.3 months (95% CI: 39.1 to 69.9)..”
Baseline demographic (Table 1), please add gender and ethnicity structure. Further, in some categories the "%" is missing, and again I recommend to put to the dosage of cortisterois exact type eg. Dexamethasone)
RESPONSE:
Thank you for your comment. We did not formally collect ethnicity information, however most of the patients were of Chinese ethnicity, per the population composition of Singapore. The gender information has been added into Table 1 (41 male, 55%). The percentages have been added. The dosing was indeed with respect to dexamethasone; we have amended the wordings in Table 1.
Generally, for the descriptive values, there is the best with median to present ranges (min-max), quartilles are possible, but very difficult for comparison with other published data.
RESPONSE:
Thank you for your comment. We have replaced the IQR with ranges instead. The paragraph (Lines 181-182) now read: “The median age of the patients in the final cohort was 61 years of age (range: 21 to 82), with a median KPS of 80 (range: 20 to 100).” Table 1 has been updated accordingly as well.
Round 2
Reviewer 2 Report
All points raised are appropriately addressed in this revised version.